# Structure of a bacterial ATP synthase

Hui Guo[1,2], Toshiharu Suzuki[3,4], John L Rubinstein[1,2,5]*

[1]The Hospital for Sick Children Research Institute, Toronto, Canada; [2]Department of Medical Biophysics, The University of Toronto, Toronto, Canada; [3]Laboratory for Chemistry and Life Science, Institute of Innovative Research, Tokyo Institute of Technology, Yokohama, Japan; [4]Department of Molecular Bioscience, Kyoto-Sangyo University, Kyoto, Japan; [5]Department of Biochemistry, The University of Toronto, Toronto, Canada

**Abstract** ATP synthases produce ATP from ADP and inorganic phosphate with energy from a transmembrane proton motive force. Bacterial ATP synthases have been studied extensively because they are the simplest form of the enzyme and because of the relative ease of genetic manipulation of these complexes. We expressed the *Bacillus* PS3 ATP synthase in *Eschericia coli*, purified it, and imaged it by cryo-EM, allowing us to build atomic models of the complex in three rotational states. The position of subunit $\varepsilon$ shows how it is able to inhibit ATP hydrolysis while allowing ATP synthesis. The architecture of the membrane region shows how the simple bacterial ATP synthase is able to perform the same core functions as the equivalent, but more complicated, mitochondrial complex. The structures reveal the path of transmembrane proton translocation and provide a model for understanding decades of biochemical analysis interrogating the roles of specific residues in the enzyme.

DOI: https://doi.org/10.7554/eLife.43128.001

## Introduction

Adenosine triphosphate (ATP) synthases are multi-subunit protein complexes that use an electro-chemical proton motive force across a membrane to make the cell's supply of ATP from adenosine diphosphate (ADP) and inorganic phosphate (Pi). These enzymes are found in bacteria and chloroplasts as monomers, and in mitochondria as rows of dimers that bend the inner membrane to facilitate formation of the mitochondrial cristae (*Davies et al., 2012*; *Paumard et al., 2002*). Proton translocation across the membrane-embedded $F_O$ region of the complex occurs via two offset half-channels (*Vik and Antonio, 1994*; *Junge et al., 1997*). Studies with *Bacillus* PS3 ATP synthase in liposomes showed that proton translocation may be driven by $\Delta$pH or $\Delta\Psi$ alone (*Soga et al., 2012*). The passage of protons causes rotation of a rotor subcomplex, inducing conformational change in the catalytic $F_1$ region to produce ATP (*Walker, 2013*) while a peripheral stalk subcomplex holds the $F_1$ region stationary relative to the spinning rotor during catalysis. For the mitochondrial enzyme, X-ray crystallography has been used to determine structures of the soluble $F_1$ region (*Abrahams et al., 1994*), partial structures of the peripheral stalk subcomplex alone (*Dickson et al., 2006*) and with the $F_1$ region (*Rees et al., 2009*), and structures of the $F_1$ region with the membrane-embedded ring of *c*-subunits attached (*Stock et al., 1999*; *Watt et al., 2010*). Recent breakthroughs in electron cryomicroscopy (cryo-EM) allowed the structures of the membrane-embedded $F_O$ regions from mitochondrial and chloroplast ATP synthases to be determined to near-atomic resolutions (*Guo et al., 2017*; *Klusch et al., 2017*; *Srivastava et al., 2018*; *Hahn et al., 2018*).

Compared to their mitochondrial counterparts, bacterial ATP synthases have a simpler subunit composition. The $F_1$ region consists of subunits $\alpha_3\beta_3\gamma\delta\varepsilon$, while the $F_O$ region is usually formed by three subunits with the stoichiometry $ab_2c_{9-15}$. Chloroplasts and a few bacteria, such as *Paracoccus denitrificans*, possess two different but homologous copies of subunit *b*, named subunits *b* and *b'*

*For correspondence: john.rubinstein@utoronto.ca

Competing interests: The authors declare that no competing interests exist.

(*Walker, 2013*). Each copy of subunit $\alpha$ and $\beta$ contains a nucleotide binding site. The non-catalytic $\alpha$ subunits each bind to a magnesium ion ($Mg^{2+}$) and a nucleotide, while the catalytic $\beta$ subunits can adopt different conformations and bind to Mg-ADP ($\beta_{DP}$), Mg-ATP ($\beta_{TP}$), or remain empty ($\beta_E$). Crystal structures of bacterial $F_1$-ATPases and $c$-rings from the $F_O$ regions of several species have been determined (*Stocker et al., 2007*; *Cingolani and Duncan, 2011*; *Morales-Rios et al., 2015*; *Shirakihara et al., 2015*; *Ferguson et al., 2016*; *Pogoryelov et al., 2009*; *Preiss et al., 2010*; *Preiss et al., 2013*; *Preiss et al., 2015*). Structures of intact ATP synthases from *E. coli* have been determined to overall resolutions of 6 to 7 Å by cryo-EM, with the $F_O$ region showing lower quality than the rest of the maps, presumably due to conformational flexibility (*Sobti et al., 2016*). In structures of both intact ATP synthase (*Sobti et al., 2016*) and dissociated $F_1$-ATPase (*Cingolani and Duncan, 2011*; *Shirakihara et al., 2015*) from bacteria, subunit $\varepsilon$ adopts an 'up' conformation that inhibits the ATP hydrolysis by the enzyme. In the thermophilic bacterium *Bacillus* PS3, this subunit $\varepsilon$ mediated inhibition is dependent on the concentration of free ATP (*Kato et al., 1997*; *Suzuki et al., 2003*; *Saita et al., 2010*). Low ATP concentrations (e.g. <0.7 mM) promote the inhibitory *up* conformation while a permissive 'down' conformation can be induced by a high concentration of ATP (e.g. >1 mM). This mechanism would allow the *Bacillus* PS3 ATP synthase to run in reverse, establishing a proton motive force by ATP hydrolysis, when the ATP concentration is sufficient to do so without depleting the cell's supply of ATP. In *E. coli*, however, in the absence of a sufficient proton motive force to drive ATP synthesis, inhibition of ATP hydrolysis by subunit $\varepsilon$ persists even when the concentration of free ATP is high (*Laget and Smith, 1979*; *Sekiya et al., 2010*).

Although bacterial ATP synthases have been subjected to extensive biochemical analysis, high-resolution structural information is lacking for the intact enzyme or the membrane-embedded proton-conducting subunit $a$ and the associated subunit $b$. We determined structures of intact ATP synthase from *Bacillus* PS3 in three rotational states by cryo-EM. The structures reached overall resolutions of 3.0, 3.0, and 3.2 Å (*Figure 1*), allowing construction of nearly complete atomic models for the entire complex. The structures reveal how loops in subunit $a$ of the bacterial enzyme fill the role of additional subunits in the $F_O$ region of the mitochondrial enzyme. Most significantly, the structures provide a framework for understanding decades of mutagenesis experiments designed to probe the mechanism of ATP synthases.

## Results and discussion

### Structure determination and overall architecture

Subunits of *Bacillus* PS3 ATP synthase, including subunit $\beta$ bearing an N-terminal 10 × His tag, were expressed from a plasmid in *E. coli* strain DK8, which lacks endogenous ATP synthase (*Klionsky et al., 1984*; *Suzuki et al., 2002*). The complex was extracted from membranes with detergent, purified by metal-affinity chromatography, and subjected to cryo-EM analysis (*Figure 1—figure supplement 1*). Three conformations corresponding to different rotational states of the enzyme were identified by ab-initio 3D classification and refined to high resolution. The 3D classes contain 45, 35, and 20% of particle images and the overall resolutions of the corresponding cryo-EM maps were 3.0, 3.0, and 3.2 Å, respectively (*Figure 1—figure supplements 2* and *3*). Estimation of local resolution suggests that the $F_1$ regions of the maps, which are larger than the $F_O$ regions and appear to dominate the image alignment process, are mostly at between 2.5 and 3.5 Å resolution, whereas the $F_O$ regions were limited to lower resolution (*Figure 1—figure supplement 3*). Focused refinement (*Bai et al., 2015*) of the $F_O$ region and peripheral stalk subunits $ab_2c_{10}$ and $\delta$ (corresponding to the subunit *OSCP* in mitochondrial ATP synthase) improved the resolution of the $F_O$ regions considerably for all three classes but not enough to resolve density for most of the amino acid side chains. An improved map of the $F_O$ region was obtained by focused refinement of the membrane-embedded region only, excluding the soluble portion of subunit $b$ with particle images from all three classes (*Figure 1—figure supplement 2*). Overall, amino acid side chain detail can be seen for subunits $\alpha_3$, $\beta_3$, $\gamma$, $\delta$, $\varepsilon$, $a$, $c_{10}$-ring, and the transmembrane $\alpha$-helices of $b_2$ (*Figure 1—figure supplement 4*). The soluble region of the two $b$-subunits was modeled as poly-alanine (*Supplementary file 1*).

The general architecture of the enzyme resembles *E. coli* ATP synthase (*Sobti et al., 2016*) and the more distantly related spinach chloroplast enzyme (*Hahn et al., 2018*) but with striking

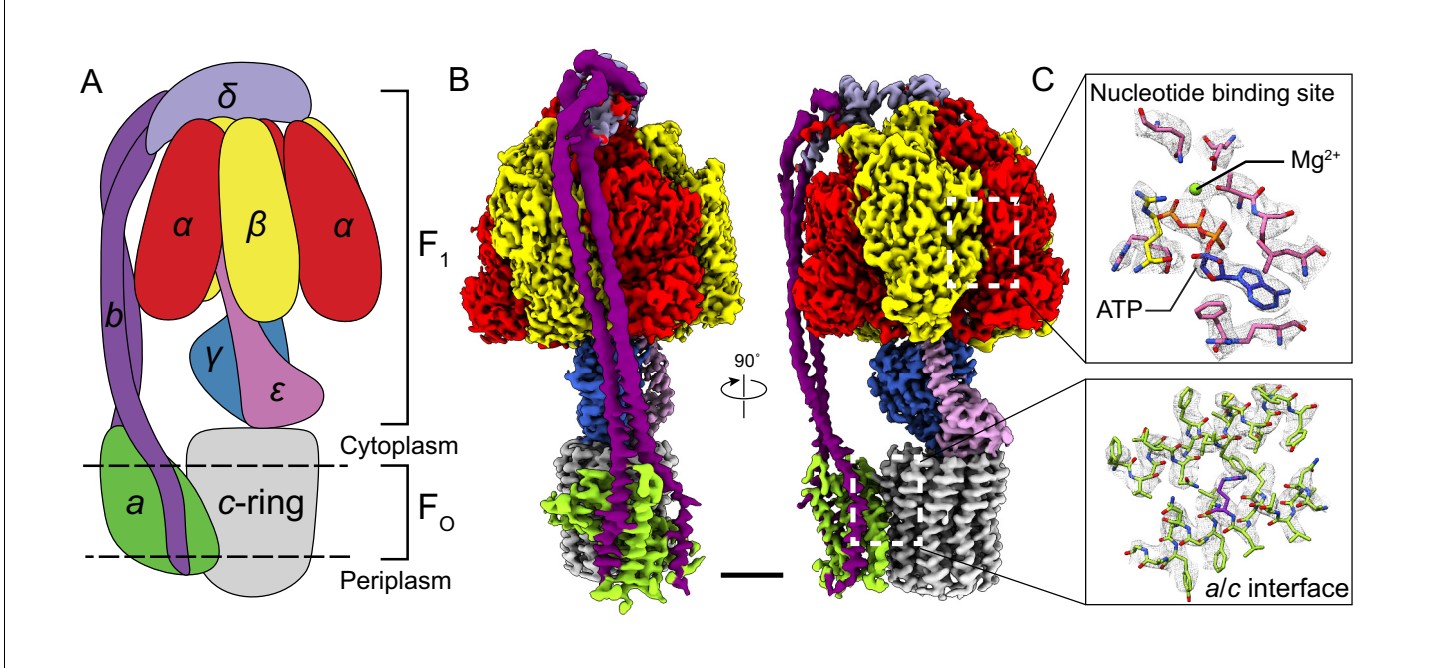

**Figure 1.** Overall structure of *Bacillus* PS3 ATP synthase. (**A**) Cartoon of ATP synthase. (**B**) Cryo-EM map of ATP synthase with subunits coloured the same as the cartoon. (**C**) Example map density that allowed construction of an atomic model. Scale bar, 30 Å.

DOI: https://doi.org/10.7554/eLife.43128.002

The following figure supplements are available for figure 1:

**Figure supplement 1.** *Bacillus* PS3 ATP synthase purification.
DOI: https://doi.org/10.7554/eLife.43128.003

**Figure supplement 2.** Cryo-EM image processing.
DOI: https://doi.org/10.7554/eLife.43128.004

**Figure supplement 3.** FSC, orientation distribution, and local resolution of the cryo-EM maps used to build atomic models.
DOI: https://doi.org/10.7554/eLife.43128.005

**Figure supplement 4.** Examples of atomic models from subunits built in the experimental cryo-EM maps.
DOI: https://doi.org/10.7554/eLife.43128.006

**Figure supplement 5.** Comparison of *Bacillus* PS3 subunits $ab_2$ with corresponding structures from spinach chloroplast and *E. coli*.
DOI: https://doi.org/10.7554/eLife.43128.007

differences. As observed previously in a *Bacillus* PS3 $F_1$-ATPase crystal structure (PDB 4XD7) (*Shirakihara et al., 2015*), the three catalytic $β$ subunits adopt 'open', 'closed', and 'open' conformations, different from the 'half-closed', 'closed', and 'open' conformations seen in the auto-inhibited *E. coli* $F_1$-ATPase (*Cingolani and Duncan, 2011*), and the 'closed', 'closed', and 'open' conformations seen in chloroplast ATP synthase (*Hahn et al., 2018*) and most mitochondrial ATP synthase structures (*Abrahams et al., 1994*; *Stock et al., 1999*). This difference, with the *half-closed* $β_{DP}$ of the *E. coli* enzyme appearing as *open* in the *Bacillus* PS3 enzyme, suggests species-specific differences in inhibition by subunit $ε$ (*Figure 1B*, pink density), which inserts into the $α/β$ interface and forces $β_{DP}$ into the *open* conformation.

Thermophilic proteins achieve stability at high temperature through mechanisms that include an increased number of ionic interactions, shorter loops between secondary structure elements, and tighter packing of hydrophobic regions (*Jaenicke and Böhm, 1998*; *Kumar and Nussinov, 2001*; *Szilágyi and Závodszky, 2000*). Comparison of individual subunit structures from the $F_1$ regions of ATP synthases from thermophiles (*Bacillus PS3* and *Caldalaklibacillus thermarum* [PDB 5HKK] (*Ferguson et al., 2016*)) and mesophiles (*E. coli* [PDB 3OAA] (*Cingolani and Duncan, 2011*), *Paracoccus denitrificans* [5DN6] (*Morales-Rios et al., 2015*), and *Spinacia oleracea* chloroplast [PDB 6FKF] (*Hahn et al., 2018*)) did not show clear evidence of tighter packing or shorter loops in the complexes from thermophiles. However, there are more ionic interactions in the $F_1$-ATPase

structures from thermophiles than from mesophiles, suggesting that these interactions may play a role in stabilizing the complexes.

In the $F_O$ region, one copy of subunit *b* is positioned at a location equivalent to that of the mitochondrial subunit *b*, while the second copy occupies the position of yeast subunit *8* (mammalian *A6L*) on the other side of subunit *a* (*Figure 1B*). Despite the different *c*-ring sizes (10 *c*-subunits in *Bacillus* PS3 versus 14 in spinach chloroplasts), the backbone positions of subunits $ab_2$ from *Bacillus* PS3 overlap with subunits *abb'* from spinach chloroplast ATP synthase (*Hahn et al., 2018*) (*Figure 1—figure supplement 5A*). Comparison of the atomic model of the $F_O$ region from *Bacillus* PS3 and the backbone model of the *E. coli* complex from cryo-EM at ~7 Å resolution (PDB 5T4O) (*Sobti et al., 2016*) showed significant structural differences in transmembrane α-helices of subunit *b* relative to subunit *a* (*Figure 1—figure supplement 5B*). Rather than reflecting true differences between *E. coli* and *Bacillus* PS3 ATP synthase structures, these deviations are likely due to the lower resolution of the *E. coli* maps.

## Flexibility in the peripheral and central stalks

As expected, the most striking difference between the three rotational states of the *Bacillus* PS3 structure is the angular position of the rotor (subunits $γεc_{10}$) (*Figure 2A*, *Video 1*). The structure of the ATP synthase, with three $αβ$ pairs in the $F_1$ region and 10 *c*-subunits in the $F_O$ region, results in symmetry mismatch between the 120° steps of the $F_1$ motor and 36° steps of the $F_O$ motor. The 120° steps of the $F_1$ motor gives an average rotational step of 3.3 *c*-subunits, with the closest integer steps being 3, 4 and 3 *c*-subunits. By comparing the positions of equivalent *c*-subunits in different rotational states, the observed rotational step sizes in the three rotational states of the ATP synthase appear to be almost exactly 3, 4 and 3 *c*-subunits (*Figure 2B*). At the present resolution, the structures of subunit *a* and the *c*-ring do not appear to differ between rotary states. Similar integer step sizes were found in yeast ATP synthase (*Vinothkumar et al., 2016*) and V-ATPase (*Zhao et al., 2015*), which also contain 10 *c*-subunits. However, non-integer steps were seen in the chloroplast (14 *c*-subunits) (*Hahn et al., 2018*) and bovine (8 *c*-subunits) (*Zhou et al., 2015*) ATP synthases, indicating that the *c*-subunit steps between the rotational states of rotary ATPases likely depends on the number of *c*-subunits.

The unequal number of *c*-subunit steps between rotational states or the different interactions made by the three $αβ$ pairs with the $b_2δ$ peripheral stalk could lead to a variable rotation speed for the *c*-ring in the active enzyme, analogous to kinetic limping in kinesin motors (*Asbury et al., 2003*). Alternatively, flexibility in the enzyme could maintain a constant rotational velocity. Indeed, flexibility is thought to be important for the smooth transmission of power between the $F_1$ and $F_O$ regions, which often have mismatched symmetries (*Wang and Oster, 1998*; *Pänke et al., 2001*; *Mitome et al., 2004*). Earlier studies suggested that the central stalk (subunits *γ* and *ε* in bacteria) is the main region responsible for the transient storage of torsional energy in rotary ATPases (*Sielaff et al., 2008*; *Wächter et al., 2011*; *Ernst et al., 2012*; *Okazaki and Hummer, 2015*). Comparison of the three rotational states of the *Bacillus* PS3 enzyme also shows that C-terminal water-soluble part of subunit *b* displays the most significant conformational variability between states, while the subunits in the $F_1$ region show little flexibility beyond the catalytic states of the $αβ$ pairs (*Figure 2C*; *Video 1*). The structure of the yeast ATP synthase $F_O$ dimer (*Guo et al., 2017*), which lacked the the $F_1$ region and an intact peripheral stalk, showed that the *c*-ring and subunit *a* are held together by hydrophobic interactions rather than by the peripheral stalk. In *Bacillus* PS3 ATP synthase, the peripheral stalk is structurally simpler and more flexible than in yeast mitochondria (*Srivastava et al., 2018*), suggesting that the bacterial subunits *a* and the *c*-ring are also held together by hydrophobic interactions and not the peripheral stalk. Given that these structures represent resting states of the bacterial ATP synthase, additional subunits, such as those in the central stalk, may show flexibility while under strain during rotation.

## Nucleotide binding in the $F_1$ region and inhibition by subunit ε

The structure of the $F_1$ region of the intact *Bacillus* PS3 ATP synthase and the earlier crystal structure of the dissociated $F_1$-ATPase (PDB 4XD7) (*Shirakihara et al., 2015*) both show that the three catalytic *β*-subunits ($β_E$, $β_{TP}$, and $β_{DP}$) adopt 'open', 'closed', and 'open' conformations, respectively (*Figure 3A*). In the crystal structure, which was prepared in the presence of CyDTA (trans-1,2-

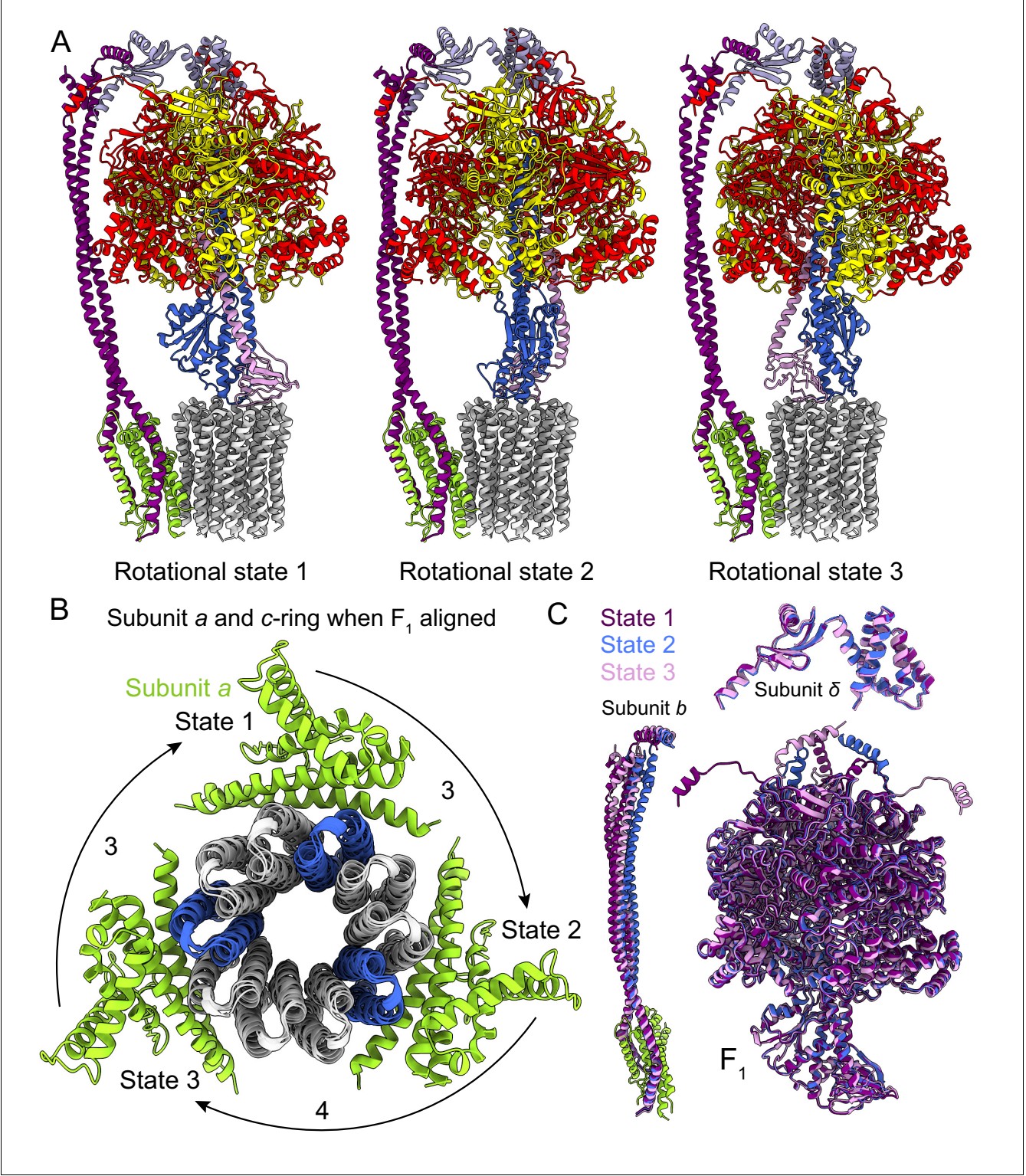

**Figure 2.** Rotational states of ATP synthase. (**A**) Atomic models of the three rotational states of *Bacillus* PS3 ATP synthase with subunits coloured the same as in *Figure 1*. (**B**) Top view of the *c*-ring and subunit *a* of the three rotational states from the cytoplasm when the F$_1$ regions of the three states are aligned. Rotation steps of the complex between states are ~3, 4, and 3 *c*-subunits. (**C**) Comparison of the atomic models of subunits *b*, δ, and other F$_1$ region subunits in the different rotational states. The *b* subunits appear to be the most flexible part of the enzyme.
DOI: https://doi.org/10.7554/eLife.43128.008

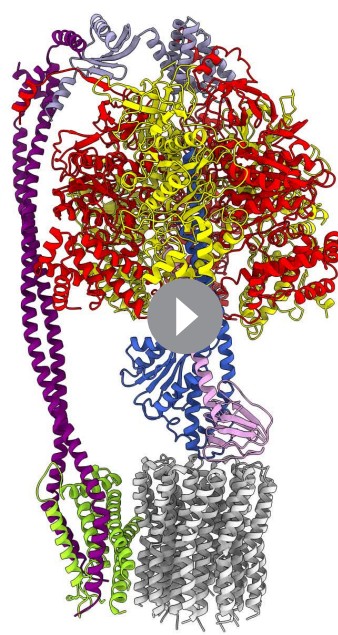

**Video 1.** Atomic models of the *Bacillus* PS3 ATP synthase in three rotational states.
DOI: https://doi.org/10.7554/eLife.43128.009

Diaminocyclohexane-N, N, N′, N′-tetraacetic acid monohydrate) as a chelating agent, there was no nucleotide in the three noncatalytic sites of the three α-subunits and the only nucleotide in a catalytic site was an ADP molecule without a $Mg^{2+}$ ion in the $\beta_{TP}$ site. In contrast, all three non-catalytic sites in the cryo-EM map are occupied by Mg-ATP, while a Mg-ADP molecule and a weak density tentatively assigned to phosphate are found in the $\beta_{TP}$ site and by the p-loop of $\beta_E$, respectively. The presence of physiological $Mg^{2+}$ ions and nucleotide occupancy (*Nalin and Cross, 1982*) in the cryo-EM map suggest that it shows a snapshot of the enzyme in the middle of its physiological catalytic cycle.

*Bacillus* PS3 ATP synthase is found in a conformation that has been proposed to allow ATP synthesis while ATP hydrolysis is auto-inhibited. In this state, subunit ε maintains an *up* conformation and inserts into the $\alpha_{DP}\beta_{DP}$ interface, forcing $\beta_{DP}$ to adopt an open conformation (*Figure 3A*, lower, dashed box) (*Shirakihara et al., 2015*). In the crystal structure (PDB 4XD7) (*Shirakihara et al., 2015*), the C-terminal sequence of subunit ε was modeled as two α-helical segments broken at Ser 106, while the cryo-EM structures show the C-terminal part is in fact entirely α-helical. In comparison, subunit ε

from the auto-inhibited *E. coli* $F_1$-ATPase structure (PDB 3OAA) (*Cingolani and Duncan, 2011*) maintains its two C-terminal α-helices (*Figure 3B*), with its $\beta_{DP}$ adopting a *half-closed* conformation that binds to Mg-ADP. The C-terminal α-helix of the *E. coli* subunit ε inserts slightly deeper into the $\alpha_{DP}\beta_{DP}$ interface but overall in a manner similar to that of the *Bacillus* PS3 subunit ε. However, the second α-helix in *E. coli* is offset by a 10-residue loop that allows it to interact with subunit γ. This interaction (*Figure 3B*, lower, dashed box) may stabilize the *up* conformation of subunit ε in *E. coli*, explaining why auto-inhibition in *E. coli* does not depend on ATP concentration (*Laget and Smith, 1979*; *Sekiya et al., 2010*) while in *Bacillus* PS3 it does. Interestingly, during ATP synthesis, *Bacillus* PS3 subunit ε was proposed to maintain the *up* conformation (*Suzuki et al., 2003*), suggesting that it only blocks ATP hydrolysis but not ATP synthesis. For a canonical ATP synthase, the substrates ADP and Pi bind to an *open* $\beta_E$. The $\beta_E$ subsequently transitions to become the *closed* $\beta_{DP}$ and then $\beta_{TP}$, driven by rotation of the central rotor, producing an ATP molecule that is ultimately released when the *closed* $\beta_{TP}$ converts back to an open $\beta_E$ (*Abrahams et al., 1994*). For the *Bacillus* PS3 ATP synthase to produce ATP with subunit ε in the *up* conformation, substrate would need to bind to the $\beta_{DP}$ site instead of the usual $\beta_E$ site, with an ATP molecule produced on transition to a *closed* $\beta_{TP}$. The cryo-EM maps show that a clash between subunit ε and $\beta_{TP}$ blocks the central rotor turning in the direction of ATP hydrolysis while it is still free to turn in the direction of ATP synthesis (*Figure 3C*), which could explain the ability of subunit ε to selectively inhibit ATP hydrolysis (*Suzuki et al., 2003*).

## Subunit organization in the $F_O$ region

In the bacterial ATP synthase structure, the $F_O$ subunits $ab_2$ display an organization similar to the yeast $F_O$ complex (PDB 6B2Z, *Figure 4A*) (*Guo et al., 2017*). Subunit *a* and the first copy of subunit *b* occupy the same positions as their yeast counterparts, while the second copy of subunit *b* is found at a position equivalent to subunit 8 in the yeast enzyme, which is known as *A6L* in mammals. Atomic models for ATP synthase from mitochondria (*Guo et al., 2017*; *Klusch et al., 2017*; *Srivastava et al., 2018*) and chloroplasts (*Hahn et al., 2018*) support the idea that transmembrane proton translocation in ATP synthases occurs via two offset half-channels formed by subunit *a*

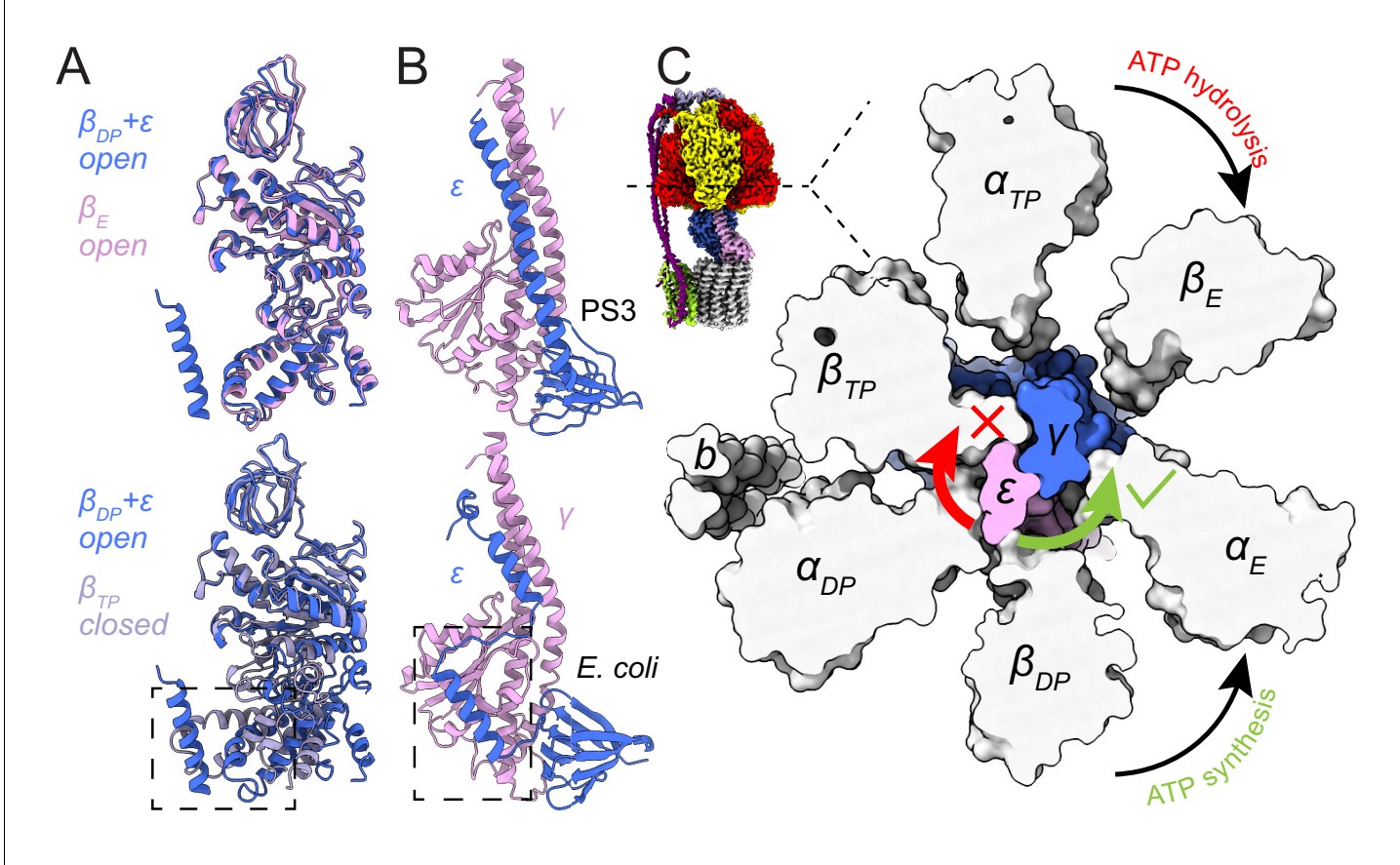

**Figure 3.** Inhibition of ATP hydrolysis by subunit $\varepsilon$. (**A**) Comparison of $\beta_{DP}$ (blue) with $\beta_E$ (pink, top) and $\beta_{TP}$ (light purple, bottom). $\beta_{DP}$ is forced to adopt an *open* conformation by subunit $\varepsilon$ (dashed box). (**B**) Comparison of subunits $\gamma$ (pink) and $\varepsilon$ (blue) of ATP synthases from *Bacillus* PS3 (top) and *E. coli* (bottom, PDB 3OAA (*Cingolani and Duncan, 2011*)). The dashed box shows additional interaction between subunits $\varepsilon$ and $\gamma$ in the *E. coli* complex. (**C**) Cross-section through the catalytic $F_1$ region of the *Bacillus* PS3 ATP synthase. Subunit $\varepsilon$ (pink) in the rotor is blocked from rotating in the direction of ATP hydrolysis (clockwise) by $\beta_{TP}$ but is free to rotate in the direction of ATP synthesis (counterclockwise).

DOI: https://doi.org/10.7554/eLife.43128.010

(*Vik and Antonio, 1994*; *Junge et al., 1997*). Subunit *a* from *Bacillus* PS3 shares 21.0% and 29.1% sequence identity with its yeast and chloroplast homologs, respectively, and the atomic model shows that the folding of these homologs is mostly conserved (*Figure 4—figure supplement 1*). Multi-sequence alignment of subunit *a* from different species indicates that bacterial and chloroplast sub-unit *a* contain a larger periplasmic loop between α-helices 3 and 4 than found in the mitochondrial subunit (*Figure 4A*, left; *Figure 4—figure supplement 2*). The sequence for this loop varies signifi-cantly among species, suggesting that it is unlikely to be involved in the core function of proton translocation, despite being proximal to the periplasmic proton half-channel. Yeast and mammalian mitochondrial ATP synthases contain subunit *f*, which has a transmembrane α-helix adjacent to the transmembrane α-helix 1 of subunit *a* (*Figure 4A*, right), anchoring subunit *b* between α-helices 5 and 6 of subunit *a*. The location of the loop between α-helices 3 and 4 of the *Bacillus* PS3 subunit *a* suggests that it serves a similar structural role, compensating for the lack of subunit *f* in bacteria. The loop forms an additional interface with subunit *b* near the periplasmic side of the membrane region and may interact with the N terminus of subunit *b* in the periplasm as well. Two interfaces are also present between the second copy of subunit *b* and subunit *a*, one with the first transmembrane α-helix, and the other with the hairpin of α-helices 3 and 4 (*Figure 4A*). The structure suggests that two interfaces are necessary for subunits *a* and *b* to maintain a stable interaction.

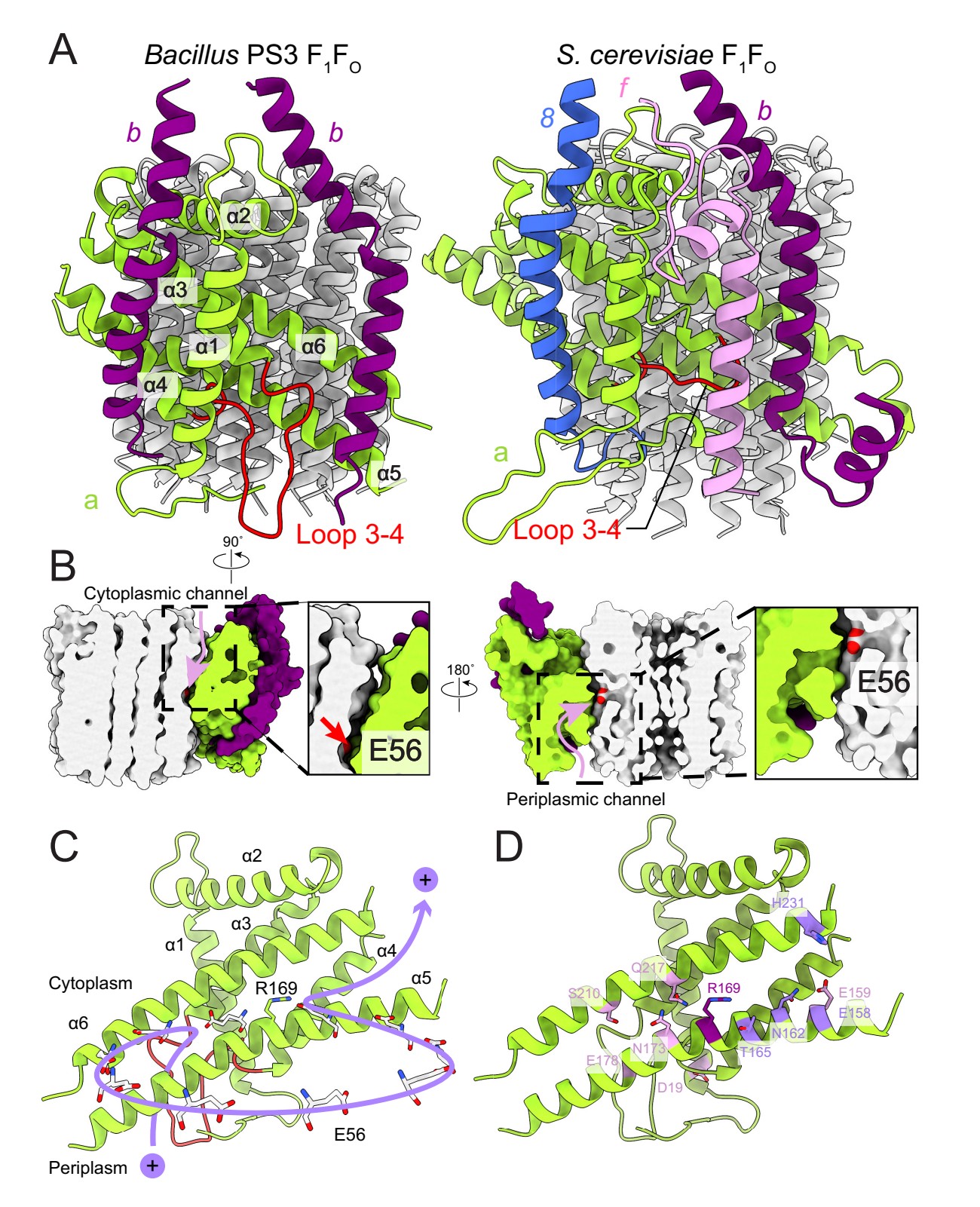

**Figure 4.** $F_O$ region subunits and proton translocation in *Bacillus* PS3 ATP synthase. (**A**) Comparison of the $F_O$ regions from *Bacillus* PS3 (left) and *S. cerevisiae* (right). (**B**) Cross sections through a surface representation of the $F_O$ region (simulated with rolling of a 1.4 Å sphere (*Goddard et al., 2018*)) show the cytoplasmic (left) and periplasmic (right) proton half-channels. (**C**) Proton translocation pathway of *Bacillus* PS3 ATP synthase. During ATP synthesis, a proton enters the complex via the periplasmic half-channel, passing between α-helices 5 and 6 of subunit *a* to bind to the Glu 56 residue of

*Figure 4 continued on next page*

*Figure 4 continued*

a subunit *c*. The proton then rotates with the *c*-ring until it reaches the cytoplasmic half-channel formed between subunit *a* and the *c*-ring. In the cytoplasmic half-channel, the proton is released from the Glu residue due to its interaction with the positively charged Arg 169 of subunit *a*. A Glu 56 residue from each protomer of the *c*-ring is shown. (D) Subunit *a* of *Bacillus* PS3 ATP synthase. Arg 169 is in purple, important residues for proton translocation identified by mutagenesis in *E. coli* ATP synthase are in pink, and other residues that appears to contribute to proton transfer in the cytosolic proton half-channel are in light purple.

DOI: https://doi.org/10.7554/eLife.43128.011

The following figure supplements are available for figure 4:

**Figure supplement 1.** Comparison of subunit *a* structures from different organisms.

DOI: https://doi.org/10.7554/eLife.43128.012

**Figure supplement 2.** Multiple sequence alignment of subunit *a*.

DOI: https://doi.org/10.7554/eLife.43128.013

**Figure supplement 3.** Positions of subunit *b* Tyr 13 and subunit *a* Gly 188.

DOI: https://doi.org/10.7554/eLife.43128.014

## Proton translocation through the $F_O$ region

The *Bacillus* PS3 ATP synthase structure implies a path for proton translocation through the bacterial complex involving two half-channels similar to the paths described for the mitochondrial and chloroplast enzymes. The cytoplasmic half-channel consists of an aqueous cavity at the interface of subunit *a* and the *c*-ring (*Figure 4B*, left). The periplasmic half-channel is formed from a cavity between α-helices 1, 3, 4 and 5 of subunit *a*, and reaches the *c*-ring via a gap between α-helices 5 and 6 (*Figure 4B*, right). In the atomic model, both channels are visible when modeling the surface with a 1.4 Å sphere that mimics a water molecule (*Goddard et al., 2018*) (*Figure 4B*). The channels are wide and hydrophilic, suggesting that water molecules could pass freely through each of the channels before accessing the conserved Glu 56 of the *c*-subunits. During ATP synthesis, protons travel to the middle of the *c*-ring via the periplasmic half-channel and bind to the Glu 56 residue of a subunit *c* (*Figure 4C*). Protonation of the glutamate allows rotation of the ring counter-clockwise, when viewed from $F_1$ toward $F_O$, delivering the subunit *c* into the hydrophobic lipid bilayer. Protonation of the remaining nine subunits in the *c*-ring returns the first glutamate to subunit *a*, now into the cytoplasmic half-channel, where it releases its proton to the cytoplasm due to interaction with the positively charged Arg 169 of subunit *a*. The proposed channels are consistent with a series of experiments probing water accessibility of residues in the *E. coli* ATP synthase subunit *a* by mutating them to cysteines and testing their accessibility by $Ag^+$ ions (*Angevine et al., 2003*; *Angevine and Fillingame, 2003*; *Angevine et al., 2007*; *Moore et al., 2008*). Residues that are close to the *c*-ring, such as S202, S206, N214, and I249 (A161, T165, N173, and G214 in *Bacillus PS3*) are among the most accessible by $Ag^+$ [52], suggesting that the half-channels do not exclude $Ag^+$. Therefore, it is likely that $Na^+$ ions, which are similar in size to $Ag^+$ ions, are also not excluded by subunit *a*. It is also known that the *c*-rings from $Na^+$- and $H^+$-driven ATP synthases have different affinities for $Na^+$ and H (*Krah et al., 2010*; *Leone et al., 2015*; *Mayer et al., 2012*; *Schlegel et al., 2012*), and that the *c*-ring of the $Na^+$-driven ATP synthase from *llyobacter tartaricus* does not bind $K^+$ or $Cs^+$ ions (*Leone et al., 2015*). Together, these results suggest that ion selectivity in ATP synthases is probably determined by the *c*-ring, not subunit *a*.

In eukaryotes, subunit *a* is encoded by the mitochondrial genome, limiting genetic interrogation of the roles of different residues. In contrast, numerous mutagenesis studies have been performed on bacterial subunits *a* and *b*, with *E. coli* ATP synthase being the most frequently studied (*Vik et al., 2000*; *Cain, 2000*). A single G9D mutation in the *E. coli* subunit *b* (positionally equivalent to Y13D in *Bacillus* PS3), results in assembled but non-functional ATP synthase (*Porter et al., 1985*), while multiple N-terminal mutations in subunit *b* can either disrupt enzyme assembly or ATP hydrolysis (*Hardy et al., 2003*). In *Bacillus* PS3, Tyr 13 is part of the transmembrane α-helix of subunit *b* and is adjacent to Gly 188 of subunit *a* (*Figure 4—figure supplement 3*, dashed box). In *E. coli* subunit *a*, Gly 188 is replaced by a leucine (Leu 229). Therefore, the G9D mutation in *E. coli* not only introduces a charged residue into a hydrophobic transmembrane α-helix, but also creates a steric clash with Leu 229 of subunit *a*, explaining why the mutation leads to an inactive enzyme. Remarkably, the single N-terminal membrane-embedded α-helix in each of the two copies of subunit *b* in the *Bacillus* PS3 ATP synthase forms different interactions with subunit *a* (*Figure 4A*). One surface interacts with

transmembrane α-helices 1, 2, 3, and 4 of subunit *a* while the other interacts with α-helices 5 and 6 and the loop between α-helices 3 and 4 of subunit *a*. Given that the N-terminal α-helix of subunit *b* makes interactions with different regions of subunit *a*, it is not surprising that mutations in this region are often detrimental to the assembly and activity of the complex. Cross-linking experiments suggested that the N terminus of the two copies of subunit *b* are in close proximity to each other (*Dmitriev et al., 1999*). However, the atomic model shows that the transmembrane α-helix of the *b*-subunits are on opposite sides of subunit *a*, suggesting that the cross-linking results may be due to non-specific interactions of *b*-subunits from neighboring ATP synthases.

In *E. coli*, Arg 210 of subunit *a* (Arg 169 in *Bacillus* PS3) tolerates the fewest mutations (*Cain and Simoni, 1989*; *Hatch et al., 1995*; *Lightowlers et al., 1987*; *Ishmukhametov et al., 2008*). Recent structures of rotary ATPases suggest that the importance of this residue derives from its role in releasing protons bound to the Glu residues of the *c*-subunits as they enter the cytoplasmic half-channel, as well as preventing short-circuiting of the proton path by protons flowing between half-channels without rotation of the *c*-ring (*Morales-Rios et al., 2015*; *Zhao et al., 2015*; *Zhou et al., 2015*; *Allegretti et al., 2015*; *Mazhab-Jafari et al., 2016*). Other residues in the *E. coli* subunit *a* identified by mutation as being functionally important include Glu 196 (Glu 159 in *Bacillus* PS3) (*Vik et al., 1988*; *Lightowlers et al., 1988*), Glu 219 (Glu 178) (*Vik et al., 1988*; *Lightowlers et al., 1988*; *Eya et al., 1991*), His 245 (Ser 210) (*Lightowlers et al., 1987*; *Cain and Simoni, 1986*; *Cain and Simoni, 1988*), Asp 44 (Asp 19) (*Howitt et al., 1990*), Asn 214 (Asn 173) (*Cain and Simoni, 1989*), and Gln 252 (Gln 217) (*Eya et al., 1991*; *Hartzog and Cain, 1993*) (*Figure 4D*). When mapped to the *Bacillus* PS3 structure, only Glu 196 (Glu 159 in *Bacillus* PS3) is close to the cytoplasmic half-channel. Extensive mutations of *E. coli* Glu 196 showed that enzyme activity depends on the charge and polarity of the residue with Glu > Asp > Gln = Ser = His > Asn > Ala > Lys (*Vik et al., 1988*). Therefore, the negative surface charge from Glu 196 (Glu 159) near the cytoplasmic half-channel facilitates proton transport across the lipid bilayer. The atomic model of subunit *a* also suggests that other residues such as *Bacillus* PS3 Thr 165, Asn 162, Glu 158, Tyr 228, and His 231, which are close to the cytoplasmic half-channel, may contribute to channel formation. Many functional residues identified by mutagenesis are clustered around the periplasmic half-channel. In the atomic model of the *Bacillus* PS3 subunit *a*, Asp 19 and Glu 178 are close to the periplasm, while Ser 210, Asn 173, and Gln 217 are deeper inside the membrane. Among these residues, Glu 178 and Ser 210 are considered to be more important to enzyme function than Asn 173 and Gln 217, as mutations of corresponding residues in *E. coli* are more likely to abolish the proton translocation by the complex (*Vik et al., 2000*). The Glu 219/His 245 residue pair in *E. coli* (*Cain and Simoni, 1988*) also occur in the *S. cerevisiae* (His 185/Glu 223) and human (His 168/Glu 203) mitochondrial ATP synthases (*Figure 4—figure supplement 2*). These residues do not appear to be close enough to form a hydrogen bond in the *S. cerevisae* $F_O$ dimer structure (*Guo et al., 2017*). In *Bacillus PS3* subunit *a*, the His residue is replaced by a serine (Ser 210) that similarly does not appear to close enough to Glu 178 to form a hydrogen bond. Interestingly, although many of these functional residues appear important, their mutation to amino acids that cannot be protonated or deprotonated often does not completely abolish proton translocation (*Vik et al., 2000*). The atomic model of *Bacillus* PS3 subunit *a* shows that the proton half-channels are wide enough for water molecules to pass through freely. This observation suggests that the function of these conserved polar and charged residues is not the direct transfer of protons during translocation. Rather, their presence may help maintain a hydrophilic environment for water-filled proton channels. This role allows different species to use unique sets of polar and charged residues to form their proton half-channels. This variability suggests a remarkably flexible proton translocation mechanism for this highly efficient macromolecular machine.

## Materials and methods

### Protein expression and purification

*E. coli* strain DK8, in which the genes encoding endogenous ATP synthase subunits were deleted (*Klionsky et al., 1984*), was transformed with plasmid pTR-ASDS (*Suzuki et al., 2002*) encoding *Bacillus* PS3 ATP synthase with a 10 × His tag at the N terminus of subunit *β*. Transformed *E. coli* cells were grown in 2 × TY medium at 37°C for 20 hr before being harvested by centrifugation at 5400 g. Cell pellets were resuspended in lysis buffer (50 mM Tris-HCl pH 7.4, 5 mM MgCl₂, 10% [w/

v] glycerol, 5 mM 6-aminocaproic acid, 5 mM benzamidine, 1 mM PMSF) and lysed with three passes through an EmulsiFlex-C3 homogenizer (Avestin) at 15 to 20 kbar. All protein preparation steps were performed at 4°C unless otherwise stated. Cell debris was removed at 12,250 g for 20 min, and the cell membrane fraction was collected by centrifugation at 184,000 g for 1 hr. Membranes were washed twice with lysis buffer before being resuspended in solubilization buffer (50 mM Tris-HCl pH 7.4, 10% [w/v] glycerol, 250 mM sucrose, 5 mM 6-aminocaproic acid, 5 mM benzamidine, 1 mM PMSF) and solubilized by the addition of glycol-diosgenin (GDN) to 1% (w/v) and mixing for 1 hr at room temperature. Insoluble material was removed by centrifugation at 184,000 g for 45 min and solubilized membranes were loaded onto a 5 ml HisTrap HP column (GE Healthcare) equilibrated with buffer A (solubilization buffer with 20 mM imidazole, 300 mM sodium chloride, and 0.02% [w/v] GDN). The column was washed with five column volumes of buffer A, and ATP synthase was eluted with three column volumes of buffer B (buffer A with 200 mM imidazole). Fractions containing ATP synthase were pooled and concentrated prior to being loaded onto a Superose 6 increase 10/300 column (GE Healthcare) equilibrated with gel filtration buffer (20 mM Tris-HCl pH 7.4, 5 mM MgCl$_2$, 10% [w/v] glycerol, 150 mM sodium chloride, 5 mM 6-aminocaproic acid, 5 mM benzamidine, 0.02% [w/v] GDN). The peak corresponding to *Bacillus* PS3 ATP synthase was pooled and concentrated to ~10 mg/ml prior to storage at −80°C.

## Cryo-EM and image analysis

Prior to grid freezing, glycerol was removed from samples with a Zeba spin desalting column (Thermo Fisher Scientific). Purified ATP synthase (2.5 µL) was applied to homemade nanofabricated EM grids (*Marr et al., 2014*) consisting of a holey layer of gold (*Russo and Passmore, 2014*; *Meyerson et al., 2014*) that had been glow-discharged in air for 2 min. Grids were then blotted on both sides in a FEI Vitrobot mark III for 26 s at 4°C and ~100% RH before freezing in a liquid ethane/propane mixture (*Tivol et al., 2008*). Cryo-EM data were collected with a Titan Krios G3 electron microscope (Thermo Fisher Scientific) operated at 300 kV equipped with a Falcon 3EC direct detector device camera automated with *EPU* software. Data were recorded as 60 s movies at 2 s per frame with an exposure rate of 0.8 electron/pixel/s, and a calibrated pixel size of 1.06 Å.

All image processing steps were performed in *cryoSPARC v2* (*Punjani et al., 2017*) unless otherwise stated. 10,940 movies were collected. Movie frames were aligned with an implementation of *alignframes_lmbfgs* within *cryoSPARC v2* (*Rubinstein and Brubaker, 2015*) and CTF parameters were estimated from the average of aligned frames with *CTFFIND4* (*Rohou and Grigorieff, 2015*). 1,866,804 single particle images were selected from the aligned frames with *Relion 2.1* (*Fernandez-Leiro and Scheres, 2017*) and beam-induced motion of individual particles corrected with an improved implementation of *alignparts_lmbfgs* within *cryoSPARC v2* (*Rubinstein and Brubaker, 2015*). A subset of 1,238,140 particle images were selected by 2D classification in *cryoSPARC v2*. After initial rounds of ab-initio 3D classification and heterogeneous refinement, three classes corresponding to three main rotational states of the enzyme were identified, containing 405,432, 314,448, and 175,694 particles images (*Figure 1—figure supplement 2*). These 3D classes were refined with non-uniform refinement to overall resolutions of 3.0 Å, 3.0 Å and 3.2 Å, respectively, with the F$_1$ region reaching higher resolution than the F$_O$ region of the complex as seen from estimation of local resolution (*Figure 1—figure supplement 3*). Masked refinement with signal subtraction (focused refinement) (*Bai et al., 2015*) around subunits $ab_2c_{10}\delta$ excluding the detergent micelle improved the map quality of the membrane-embedded region as well as the peripheral stalk for all three classes. The membrane-embedded region (subunits $ac_{10}$ and transmembrane α-helices of the *b*-subunits) was improved further by focused refinement with particle images from all three classes, yielding a map at 3.3 Å resolution. All Fourier shell correlation (FSC) curves were calculated with independently refined half-maps and resolution was assessed at the 0.143 criterion with correction for the effects of masking maps. For illustration purposes, composite maps for each of the three rotational states were generated by combining the F$_1$ region of the maps from non-uniform refinement, the peripheral stalk region from the maps obtained with focused refinement of subunits $ab_2c_{10}\delta$, and the map from focused refinement of the membrane-embedded region. Specifically, each map was multiplied by a mask surrounding the region of interest and the resulting maps were adjusted to similar absolute grey scale by multiplying with a constant with *relion_image_handler* before being merged with the maximum function volume operation in *UCSF Chimera* (*Pettersen et al., 2004*). These composite maps were not used for model refinement.

## Model building and refinement

Atomic models for subunits $\alpha_3\beta_3\gamma\varepsilon\delta$ from all three rotational states were built with *Coot* (*Emsley and Cowtan, 2004*) into the maps of the intact complex from non-uniform refinement using PDB 4XD7 (*Shirakihara et al., 2015*) and PDB 6FKF (*Hahn et al., 2018*) as initial models for subunits $\alpha_3\beta_3\gamma\varepsilon$ and subunit $\delta$, respectively. Subunits $ac_{10}$ and the membrane-embedded regions of subunits $b_2$ were built de novo in the 3.3 Å map of the membrane-embedded region of the complex from focused refinement. Backbone models of the soluble region of subunits $b_2$ for all three conformations were built with the maps from focused refinement of the peripheral stalk. Models were refined into their respective maps with *phenix.real_space_refine* (*Adams et al., 2010*) using secondary structure and geometric restraints followed by manual adjustments in *Coot* (*Supplementary file 1*). The quality of the models was evaluated by *MolProbity* (*Chen et al., 2010*) and *EMRinger* (*Barad et al., 2015*). To generate full models for all three rotational states, the model of subunits $ac_{10}$ and the membrane region of subunit $b_2$ were fit into the full maps of each conformation as three rigid bodies ($a$, $c_{10}$, and $b_2$ membrane region) with *phenix.real_space_refine*. For classes 1 and 3, the backbone models of the soluble region of subunit $b_2$ did not fit the full maps well, and thus the fit was improved by molecular dynamics flexible fitting (MDFF) (*Trabuco et al., 2008*). The final composite model for each rotational state was generated by combining the models of subunits $\alpha_3\beta_3\gamma\varepsilon\delta$, the rigid body refined subunits $ac_{10}$ and membrane region of $b_2$, and the backbone model of the soluble region of $b_2$. Figures and movie were generated with Chimera (*Pettersen et al., 2004*) and ChimeraX (*Goddard et al., 2018*).

## Acknowledgements

We thank Dr. Samir Benlekbir (the Hospital for Sick Children) for helping with cryo-EM data collection and Prof. Tomitake Tsukihara (Osaka University, Japan) for discussions. This work was supported by Canadian Institutes of Health Research operating grant MOP 81294. Cryo-EM data was collected at the Toronto High-Resolution High-Throughput cryo-EM facility, supported by the Canada Foundation for Innovation and Ontario Research Fund. HG was supported by an Ontario Graduate Scholarship and a University of Toronto Excellence Award. TS was supported by Japan Society for the Promotion of Science Grants-in-Aid for Scientific Research (KAKENHI) Grant JP18H02409. JLR was supported by the Canada Research Chairs program.

## Additional information

### Funding

| Funder | Grant reference number | Author |
| --- | --- | --- |
| Canadian Institutes of Health Research | MOP 81294 | John L Rubinstein |
| Canada Research Chairs | | John L Rubinstein |
| Japan Society for the Promotion of Science | JP18H02409 | Toshiharu Suzuki |
| Canada Foundation for Innovation | | John L Rubinstein |

The funders had no role in study design, data collection and interpretation, or the decision to submit the work for publication.

### Author contributions

Hui Guo, Conceptualization, Data curation, Formal analysis, Validation, Investigation, Visualization, Methodology, Writing—original draft, Writing—review and editing; Toshiharu Suzuki, Methodology, Writing—review and editing; John L Rubinstein, Conceptualization, Supervision, Funding acquisition, Methodology, Writing—original draft, Project administration, Writing—review and editing

Author ORCIDs
Hui Guo (ID) http://orcid.org/0000-0001-7007-2876
John L Rubinstein (ID) http://orcid.org/0000-0003-0566-2209

Decision letter and Author response
Decision letter https://doi.org/10.7554/eLife.43128.040
Author response https://doi.org/10.7554/eLife.43128.041

## Additional files

### Supplementary files

• Supplementary file 1. Cryo-EM data acquisition, processing, atomic model statistics, and map/model depositions. (**A**) Cryo-EM data acquisition and image processing. (**B**) Map and model statistics. (**C**) Residues included in atomic models. (**D**) Deposited maps and associated coordinate files.
DOI: https://doi.org/10.7554/eLife.43128.015
• Transparent reporting form
DOI: https://doi.org/10.7554/eLife.43128.016

### Data availability

CryoEM maps have been deposited in EMDB and atomic models in PDB.

The following datasets were generated:

| Author(s) | Year | Dataset title | Dataset URL | Database and Identifier |
|---|---|---|---|---|
| Guo H, Rubinstein JL | 2018 | Intact class 1 | http://www.ebi.ac.uk/pdbe/entry/emdb/EMD-9333 | Electron Microscopy Data Bank, EMD-9333 |
| Guo H, Rubinstein JL | 2018 | Intact class 1 | http://www.rcsb.org/structure/6N2Y | Protein Data Bank, 6N2Y |
| Guo H, Rubinstein JL | 2018 | Intact class 2 | http://www.ebi.ac.uk/pdbe/entry/emdb/EMD-9334 | Electron Microscopy Data Bank, EMD-9334 |
| Guo H, Rubinstein JL | 2018 | Intact class 2 | http://www.rcsb.org/structure/6N2Z | Protein Data Bank, 6N2Z |
| Guo H, Rubinstein JL | 2018 | Intact class 3 | http://www.ebi.ac.uk/pdbe/entry/emdb/EMD-9335 | Electron Microscopy Data Bank, EMD-9335 |
| Guo H, Rubinstein JL | 2018 | Intact class 3 | http://www.rcsb.org/structure/6N30 | Protein Data Bank, 6N30 |
| Guo H, Rubinstein JL | 2018 | Focused Fo/stalk class 1 | http://www.ebi.ac.uk/pdbe/entry/emdb/EMD-9336 | Electron Microscopy Data Bank, EMD-9336 |
| Guo H, Rubinstein JL | 2018 | Focused Fo/stalk class 2 | http://www.ebi.ac.uk/pdbe/entry/emdb/EMD-9337 | Electron Microscopy Data Bank, EMD-9337 |
| Guo H, Rubinstein JL | 2018 | Focused Fo/stalk class 3 | http://www.ebi.ac.uk/pdbe/entry/emdb/EMD-9338 | Electron Microscopy Data Bank, EMD-9338 |
| Guo H | 2018 | Focused Fo | http://www.rcsb.org/structure/6N2D | Protein Data Bank, 6N2D |
| Guo H, Rubinstein JL | 2018 | Focused Fo | http://www.ebi.ac.uk/pdbe/entry/emdb/EMD-9327 | Electron Microscopy Data Bank, EMD-9327 |

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
