## [Decision Letter]

Thank you for submitting your article "Structure of a bacterial ATP synthase" for consideration by *eLife*. Your article has been reviewed by Michael Marletta as the Senior Editor, a Reviewing Editor, and three reviewers. The following individuals involved in review of your submission have agreed to reveal their identity: David Mueller (Reviewer #1); Steven Vik (Reviewer #2); Wolfgang Junge (Reviewer #3).

The reviewers have discussed the reviews with one another and the Reviewing Editor has drafted this decision to help you prepare a revised submission.

Summary:

Adding to the recent progress in elucidating by cryo-EM the structure of the holoenzyme, FOF1, from different species the authors focused on heterologous expressed ATP synthase FOF1 of bacillus PS3. The resolution of the structures reached 3.0-3.1 Å, allowing a close look at several important features of the enzyme.

Their work produced three important results:

a) Species specific structural differences are related to different inhibition of ATP hydrolysis by the ε-subunit. Their cryo-EM structures of PS3-FOF1 explains why the inhibition of ATP hydrolysis in the *E. coli* enzyme does not depend on the ATP concentration while it does in PS3. Moreover, in PS3 they found steps by 3, 4, and 3 copies of subunit *c* on the ring as previously reported for the yeast ATP synthase and V-ATPase, that are not found in chloroplasts (14c) bovine mitos (8c).

b) In contrast to kinetic results they found that the water soluble part of subunit *b* displays more significant conformational flexibility between states than the central stalk. This will be interesting to follow.

c) The hypothetical model of the protonic drive of FO, with two offset half-channels lining the c-ring, is again corroborated by their structure. The authors focused on the half-channels to the conserved Glu residues of the *c* subunits. Because of the relatively high resolution of the structure, these channels were readily visible, and appeared to have unobstructed pathways for water molecules to penetrate. Polar residues that lined these pathways were discussed in the context of mutations that had been analyzed earlier, primarily in the *E. coli* enzyme.

All 3 referees agreed that the data are convincing, the manuscript makes major contributions towards understanding the function of bacterial ATP synthases and that the article is very well written.

1) It would useful if the authors compiled a table indicating which residues of each subunit appear in the model.

2) The following references on the elastic function of the rotor should be cited:

Okazaki and Hummer, (2015) and Ernst et al., 2012.

The authors may, at their discretion, consider commenting on two items:

3) The mechanism of FO as a whole is mono-specific for the proton, due to the almost full round of the bound proton on an acid residue of the c-ring. The question is whether each of the half channels is already mono-specific (this is what Mitchell coined a "proton well"). The authors found water access. Can they exclude access of say Na^+^ and K^+^?

4) Based on the structure with the fixed position of *b_2_δ* relative to (*αβ)_3_*, do the authors expect kinetic "limping" during rotary ATP hydrolysis?

5) In Figure 4C the violet, circular arrows describe the proton pathway. At the periplasmic entry the path seems to go between helices 5 and 6 of subunit *a*. Is that intended, or should it go between the *a* and c subunits? Also, it was hard to recognize that the Glu 56 residues were shown for each subunit *c*. A better description in the figure legend would help.

6) The authors did not comment on whether residues Glu 178 and Ser 210 appear to interact by Hydrogen bonding. In many species this is a Glu-His pair.

7) It would be interesting to know if the authors observed any features of the enzyme that are characteristic of thermophilic proteins, perhaps illustrating an adaption relative to other organisms.

8) Typo: in the Discussion section "a water molecule" or "water molecules".

---

## [Author Response]

1) It would useful if the authors compiled a table indicating which residues of each subunit appear in the model.

We agree that this information is important and have added it in Table 1C.

2) The following references on the elastic function of the rotor should be cited:Okazaki and Hummer, (2015) and Ernst et al., 2012.

We have cited these two important papers along with the other manuscripts that we cited earlier.

The authors may, at their discretion, consider commenting on two items:3) The mechanism of FO as a whole is mono-specific for the proton, due to the almost full round of the bound proton on an acid residue of the c-ring. The question is whether each of the half channels is already mono-specific (this is what Mitchell coined a "proton well"). The authors found water access. Can they exclude access of say Na^+^ and K^+^?

The reviewers raise an interesting point. We have not tested the ability of the *Bacillus PS3* ATP synthase to work with Na^+^ or K^+^ ions, but there is some literature that suggests that it is the *c*-ring that determines the ion selectivity of the enzyme. We have added the following text to the manuscript (subsection “Proton translocation through the FO region”):

“The proposed channels are consistent with a series of experiments probing water accessibility of residues in the *E. coli* ATP synthase subunit *a* by mutating them to cysteines and testing their accessibility by Ag^+^ ions. […] Together, these results suggest that ion selectivity in ATP synthases is probably determined by the *c*-ring, not subunit *a*.”

4) Based on the structure with the fixed position of b_2_δ relative to (αβ)_3_, do the authors expect kinetic "limping" during rotary ATP hydrolysis?

As far as we understand, “kinetic limping” is seen in kinesin motors, where the timing of consecutive steps is unequal. For the *Bacillus PS3* ATP synthase, the rotation steps of the complex appear to be 3, 4, and 3 c-subunits for each ATP molecule synthesized. Our structure, which provides static snapshots of the enzyme at rest, does not give information about the speeds of transitions between these states or even if these states exist while the enzyme is synthesizing ATP. To raise this possibility, we have added the following text to subsection “Flexibility in the peripheral and central stalks”:

“The unequal number of *c*-subunit steps between rotational states or the different interactions made by the three *αβ* pairs with the *b_2_δ* peripheral stalk could lead to a variable rotation speed for the *c-*ring in the active enzyme, analogous to kinetic limping in kinesin motors. Alternatively, flexibility in the enzyme could maintain a constant rotational velocity.”

5) In Figure 4C the violet, circular arrows describe the proton pathway. At the periplasmic entry the path seems to go between helices 5 and 6 of subunit a. Is that intended, or should it go between the a and c subunits? Also, it was hard to recognize that the Glu 56 residues were shown for each subunit c. A better description in the figure legend would help.

The proton path does indeed go between α-helices 5 and 6 of subunit *a*. We have modified this Figure to make the path clearer and modified the Figure caption with more descriptive text:

“During ATP synthesis, a proton enters the complex via the periplasmic half-channel, passing between α-helices 5 and 6 of subunit *a* to bind to the Glu 56 residue of a subunit *c*. The proton then rotates with the *c*-ring until it reaches the cytoplasmic half-channel formed between subunit *a* and the *c*-ring. In the cytoplasmic half-channel the proton is released from the Glu residue due to its interaction with the positively charged Arg 169 of subunit *a*. A Glu 56 residue from each protomer of the *c*-ring is shown.”

6) The authors did not comment on whether residues Glu 178 and Ser 210 appear to interact by Hydrogen bonding. In many species this is a Glu-His pair.

In our earlier structure of the yeast dimeric F_O_ complex (PDB 6B2Z) the distance between Glu/His pair is more than 5 Å. Although the side chain density of the cryo-EM map is relatively weak for the Glu 223 residue (a common feature for carboxyl groups in cryo-EM maps), it seems unlikely that this Glu-His pair is hydrogen bonded. In *Bacillus PS3* subunit a, we did not build the side chain for Glu 178 due to weak density. However, judging from the orientation of C_β_, Glu 178 and Ser 210 probably do not interact by hydrogen bonding either. We have added the following text to page (subsection “Proton translocation through the FO region”):

“The Glu 219/His 245 residues in *E. coli* also occur in the *S. cerevisiae* (His 185/Glu 223) and human (His 168/Glu 203) mitochondrial ATP synthases (Figure 4—figure supplement 2). These residues do not appear to be close enough to form a hydrogen bond in the *S. cerevisae* F_O_ dimer structure. In *Bacillus PS3* subunit *a*, the His residue is replaced by a serine (Ser 210) that similarly does not appear to close enough to Glu 178 to form a hydrogen bond.”

7) It would be interesting to know if the authors observed any features of the enzyme that are characteristic of thermophilic proteins, perhaps illustrating an adaption relative to other organisms.

We compared the structures of thermophilic and mesophilic ATP synthases and found that the main difference appears to be in the number of ionic interactions. We have added the following text to Subsection “Structure determination and overall architecture”:

“Thermophilic proteins achieve stability at high temperature through mechanisms that include an increased number ionic interactions, shorter loops between secondary structure elements, and tighter packing of hydrophobic regions. Comparison of individual subunit structures from the F_1_ regions of ATP synthases from thermophiles (*Bacillus PS3* and *Caldalaklibacillus thermarum* [PDB 5HKK]) and mesophiles (*E. coli* [PDB 3OAA], *Paracoccus denitrificans* [5DN6], and *Spinacia oleracea* chloroplast [PDB 6FKF]) did not show clear evidence of tighter packing or shorter loops in the thermophilic complexes. However, there are more ionic interactions, including those with distances < 4 Å and < 8 Å, in the thermophilic F_1_-ATPase structures than in the mesophiles, suggesting that these interactions may play a role in stabilizing the thermophilic complexes.”

8) Typo: in the Discussion section "a water molecule" or "water molecules".

We have corrected this typographical error.